# AerSett v1.0: A simple and straightforward model for the settling speed of big spherical atmospheric aerosol.

Sylvain Mailler[1,2], Laurent Menut[1], Arineh Cholakian[1], and Romain Pennel[1]

[1]LMD/IPSL, École Polytechnique, Institut Polytechnique de Paris, ENS, PSL Research University, Sorbonne Université, CNRS, Palaiseau France

[2]École des Ponts-ParisTech, Marne-la-Vallée, France

**Correspondence:** Sylvain Mailler (sylvain.mailler@lmd.ipsl.fr)

**Abstract.** This study introduces AerSett v1.0 (AERosol SETTling version 1.0), a model giving the settling speed of big spherical aerosols in the atmosphere without going through an iterative equation resolution. We prove that, for all spherical atmospheric aerosols with diameter $D$ up to $1000\,\mu\text{m}$, this direct and explicit method including the drag coefficient formulation of Clift and Gauvin (1971) and the Davies (1945) slip correction factor gives results within $2\%$ of the exact solution obtained from numerical resolution of a non-linear fixed-point equation. This error is acceptable considering the uncertainties on the drag coefficient formulations themselves. For $D < 100\,\mu\text{m}$, the error is below 0.5%. We provide a Fortran implementation of this simple and straightforward model, hoping that more Chemistry-Tranport models and General Circulation models will be able to take into account large-particle drag correction to the settling speed of big spherical aerosol particles in the atmosphere, without performing an iterative and time-consuming calculation.

## 1 Introduction

One of the main purposes of chemistry-transport modeling is to simulate the aerosol concentration in the atmosphere as accurately as possible. The settling velocity of aerosols is a key driver of their dry removal from the atmosphere (Zhang et al., 2001). Dry removal being the only sink for atmospheric aerosol under dry conditions, any error on representing the settling velocity of atmospheric aerosol will have a direct impact on their modelled concentrations. Fortunately, for particles with diameter $D < 10\mu m$, which are the most significant for health effects, the Stokes law (Stokes, 1851) along with slip correction factors (Cunningham, 1910; Davies, 1945) gives a straightforward and accurate way to calculate the settling speed of aerosol particles.

However, dust particles exist in the atmosphere with different sizes from diameter $D \simeq 0.1\,\mu\text{m}$ to $D > 100\,\mu\text{m}$ (Ryder et al., 2019). These authors classify mineral dust particles according into three modes as follows:

- $0.1\,\mu\text{m} < D < 2.5\,\mu\text{m}$: accumulation mode

- $2.5\,\mu\text{m} < D < 20\,\mu\text{m}$: coarse mode

- $20\,\mu\text{m} < D < \infty$: giant mode

The contribution of the giant mode is substantial, at least over the Sahara: Ryder et al. (2019) show that not taking into account giant dust particles over the Sahara results in underestimating mass concentration by 40%, and extinction by as much as 18% for shortwave radiation and 26% for longwave radiation. Dust particles with diameter up to $100\,\mu\mathrm{m}$ are present not only above the Sahara (Ryder et al., 2019) but also far away from emission sources: Betzer et al. (1988) have observed dust particles with $D > 75\,\mu\mathrm{m}$ in the atmosphere more than $10\,000\,\mathrm{km}$ away from their source. More recently, van der Does et al. (2018) have observed dust particles with diameter up to $450\,\mu\mathrm{m}$ over the Atlantic Ocean, more than $2400\,\mathrm{km}$ away from the West-African coast. Modelling coarse and giant dust particles is still a very challenging task. For example, Drakaki et al. (2022) show that the WRFV4.2.1 model with a version of the GOCART-AFWA dust scheme modified to include coarse and giant dust particles underestimates the lifetime of coarse and giant dust particles in the atmosphere. They show that their simulation results are closer to observation when they include an artificial reduction of particles' settling velocities by 60% to 80% (depending on the diameter). This reduction is a way to account for underrepresented mechanisms such as non-sphericity of particles (Mallios et al., 2020), or their electric charges, which have been discussed as possible factors explaining a longer atmospheric lifetime of coarse dust particles (Adebiyi and Kok, 2020). Several observational and modelling studies have addressed the question of coarse and giant dust particles in the atmosphere: doing an exhaustive bibliographical overview on this question falls beyond the scope of the present study. For a more complete bibliography, the reader is referred to van der Does et al. (2018), Ryder et al. (2019) and Drakaki et al. (2022).

The settling speed of giant particles deviates substantially from the Stokes law, an effect that can be taken into account using mathematical formulations known as large-particle drag corrections. Usually, these large-particle drag corrections are performed by using empirical formulations of the drag-coefficient $C_d$ as a function of the Reynolds number $Re$ (typically the one provided by Clift and Gauvin (1971)), and numerically solving an equation to obtain an estimate of the settling speed $v_\infty$ as a function of the characteristics of the particle and of ambient air. This method is robust and permits to perform studies like Drakaki et al. (2022) taking into account this effect, but incurs in large calculation costs. Since the important impact of giant dust particles on the dust concentration and optical effect has been demonstrated (e.g. Ryder et al. (2019)), there is an emerging need to solve the problems that hinder the representation of giant dust particles in CTMs and General circulation models. Designing a robust and efficient method to calculate the settling speed of giant dust particles is a step in this direction. Until now, the gravitational settling speed in most chemistry-transport models is calculated with a plain Stokes formulation and a slip correction factor for the smallest particles, as in Zhang et al. (2001), but without a large-particle drag correction (*e.g.*, Sič et al. (2015), Rémy et al. (2019), Shu et al. (2021)). An exception to this is the recent development exposed by Drakaki et al. (2022) in the GOCART-AFWA dust scheme of WRFV4.2.1. In that study, the Clift and Gauvin (1971) drag coefficient correction is taken into account by a bisection method, performed at each time step, in each model cell and for each model size bin to calculate the settling speed as a function of the particle properties and the atmospheric conditions.

Since it has been highlighted by Drakaki et al. (2022) that taking into account large-particle drag correction is important for representing the settling of giant dust particles, the goal of this article is to give a simple, robust and computationnally efficient expression to calculate the settling speed of spherical atmospheric aerosol, including large-particle drag correction. As discussed thoroughly in Goossens (2019), several parameterizations exist for the drag coefficient, each of them fitting the

reference data only in part. These parameterizations give drag coefficients which differ between them and from measurements by a few percents. Among these parameterizations, the Clift and Gauvin (1971) and Cheng (2009) formulations seem to perform better according to the objective scores presented in Table 2 of Goossens (2019). In the present study, we base our calculations on the Clift and Gauvin (1971) drag coefficient formulation, and use Cheng (2009) as a benchmark of the uncertainty that can exist between several state-of-the-art drag coefficient formulations.

In Section 2 we review the basic equations that govern the settling speed of spherical aerosol particles in the atmosphere (leaving aside slip correction); in Section 3 we lay the basis for a new methodology for finding the settling speed of the particle as a function of the particle and surrounding gas parameters. In section 4, we apply this method to the case of the drag coefficient parameterization of Clift and Gauvin (1971) and give an approximated expression of the large-particle drag correction factor. Sections 2, 3 and 4 focus on the large-particle drag correction, ignoring the slip correction factor. In section 5, we extend the results of the previous sections by showing how to combine the slip correction factor with the large-particle drag correction, and present the equations that define the AerSett method, including both effects. In Section 6, we evaluate the computational benefit of this method compared to other numerical resolution strategies, before concluding in Section 7 by summarizing the method we propose for the calculation of the settling speed of large spherical aerosol particles as well as future prospects for improving the representation of the settling speed of aerosol particles in chemistry-transport models.

## 2   Basic equations

We will follow the notations of Mallios et al. (2020). The settling speed $v_\infty$ of a spherical particle with diameter $D$ is given by:

$$\frac{1}{2}C_d A_p \rho_a v_\infty^2 = (\rho_p - \rho_a) V_p g, \tag{1}$$

Where $C_d$ is the drag coefficient, $A_p$ the projected area of the particle normal to the flow, $\rho_a$ the density of air, $\rho_p$ the density of the settling particle, $v_\infty$ the settling speed of the particle, $V_p$ the particle volume and $g$ the acceleration of gravity.

In the case of spherical particles and for extremely small Reynolds numbers, Stokes (1851) has shown that:

$$C_d = \frac{12}{Re}, \tag{2}$$

where the Reynolds number $Re$ is defined as:

$$Re = \frac{\rho_a D v_\infty}{2\mu}. \tag{3}$$

For all the calculations below, we have use the following empirical expression for dynamic viscosity $\mu$ (NOAA/NASA/USAF (1976)):

$$\mu = \frac{\beta T^{\frac{3}{2}}}{T + S}, \tag{4}$$

 with $\beta = 1.458 \times 10^{-6}\,\mathrm{kg\,s^{-1}\,m^{-1}\,K^{-\frac{1}{2}}}$ and $S = 110.4\,\mathrm{K}$.

While Eq. 2 holds only for extremely small Reynolds number ($Re < 0.1$), Clift and Gauvin (1971) has given an empirical expression of $C_d$ as a function of $Re$, which holds up to $Re = 10^5$:

$$C_d = \frac{12}{Re}\left(1 + 0.2415\,Re^{0.687}\right) + \frac{0.42}{1 + \frac{19019}{Re^{1.16}}}, \tag{5}$$

where exponents 0.687 and 1.16 and coefficients 0.2415, 0.42 and 19019 are empirical values that permit Eq. 5 to fit experimental data. Hereinafter, we will refer to Eq. 5 as the Clift-Gauvin formula.

Equations 1 and 5 are two equations for two unknowns, $v_\infty$ and $C_d$. For small Reynolds numbers ($Re < 0.1$), Eq. 5 reduces to Eq. 2 and we obtain:

$$v_\infty^{Stokes} = \frac{D^2\left(\rho_p - \rho_a\right)g}{18\mu}. \tag{6}$$

If the Reynolds number exceeds 0.1, Eq 2 does not hold, and $v_\infty$ does not have a general analytical expression. The solution to Eq. 2 can be obtained by an iterative fixed-point method as suggested in van Boxel (1998), or by a bisection method as in Drakaki et al. (2022). The results of this numerical calculation is shown in Fig. 1. Fig. 1a shows that the Stokes equation (Eq. 6) gives excellent results for $D < 20\,\mu\mathrm{m}$, but that deviations from it due to the departure of the drag coefficient from Eq. 2 gradually arise when $D$ exceeds $20\,\mu\mathrm{m}$, reaching $-30\%$ when $D \simeq 100\,\mu\mathrm{m}$, and $-90\%$ when $D \simeq 1000\,\mu\mathrm{m}$ (Fig. 1c). While particles with diameter $D \simeq 1000\,\mu\mathrm{m}$ are not a concern for chemistry-transport modelling, those with $D \simeq 100\,\mu\mathrm{m}$ are an emerging concern, due to recent observations of particles with such diameters far away from their source (van der Does et al. (2018)).

Solving Eq. 1 with an iterative method as suggested in van Boxel (1998) demands several iterations when the diameter of the particle gets close to $100\,\mu\mathrm{m}$ or beyond (see Section 6). This is why we will expose a way to estimate $v_\infty$ from the physical parameters of the problem in a straightforward way.

## 3   Expressing $v_\infty$ from the parameters of the problem

To slightly generalize matters, let us rewrite Eq. 5 as:

$$C_d = \frac{12}{Re}\left(1 + f\left(Re\right)\right), \tag{7}$$

where $f$ is a function characterizing the deviation of $C_d$ from its Stokes (1851) expression.

Injecting Eq. 7 into Eq. 1, with $A_p = \frac{\pi D^2}{4}$ and $V_p = \frac{\pi D^3}{6}$, we obtain:

$$v_\infty = \frac{D^2\left(\rho_p - \rho_a\right)g}{18\mu} \times \frac{1}{\left(1 + f\left(Re\right)\right)} \tag{8}$$

$$= v_\infty^{Stokes} \times \frac{1}{\left(1 + f\left(Re\right)\right)} \tag{9}$$

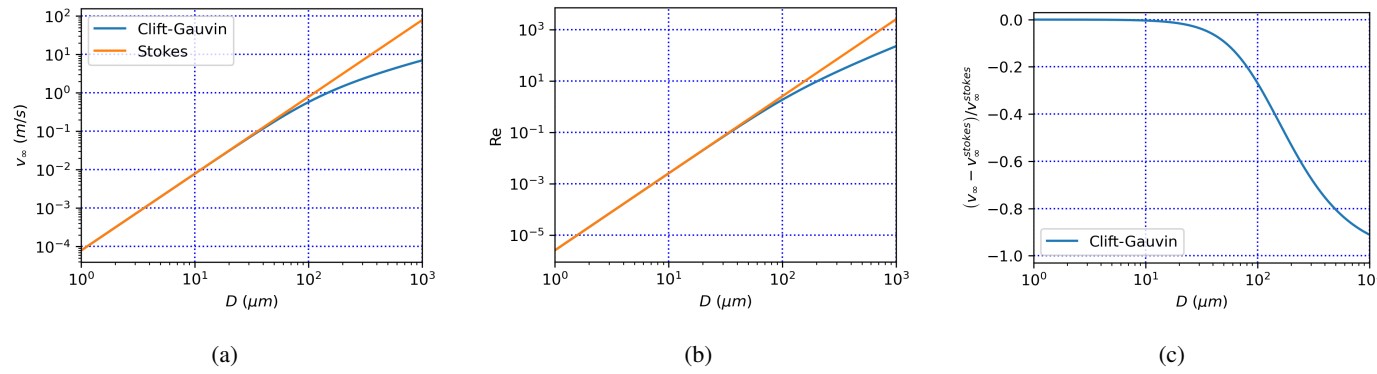

**Figure 1.** (a) settling speed of a spherical particle with $\rho = 2650\,\mathrm{kg\,m^{-3}}$ in air with $P = 101325\,\mathrm{Pa}$ and $T = 293.15\,\mathrm{K}$ as a function of diameter with $C_d$ from equations 5 (blue) and 6 (orange). (b) Reynolds number from equations 5 (blue) and 6 (orange), and (c) relative difference of $v_\infty$ from $v_\infty^{Stokes}$

Of course, Eq. 9 does not give an explicit formulation of $v_\infty$ since $Re$ depends on $v_\infty$ (Eq. 3). However, we are looking for a way to take advantage of Eq. 9 to obtain an explicit estimate of $v_\infty$, so that we introduce the deviation of $v_\infty$ from $v_\infty^{Stokes}$, $\delta$, as:

$$v_\infty = v_\infty^{Stokes}\left(1 + \delta\right) \tag{10}$$

The terminal Reynolds number of the particle is equal to:

$$Re = \frac{\rho_a D v_\infty^{Stokes}\left(1 + \delta\right)}{2\mu} = \left(1 + \delta\right)\frac{\rho_a D^3\left(\rho_p - \rho_a\right)g}{36\mu^2} \tag{11}$$

Let us introduce:

$$R = \frac{\rho_a D^3\left(\rho_p - \rho_a\right)g}{36\mu^2} \tag{12}$$

$R$ is the Reynolds number of the particle if it would have if it would be falling at speed $v_\infty^{Stokes}$: we will call it the *virtual Reynolds number*. The Reynolds number of the particle falling at its corrected settling speed $v_\infty$ is $R\left(1 + \delta\right)$. With these modifications, Eq. 9 can be rewritten as:

$$1 + \delta = \frac{1}{1 + f\left(\left(1 + \delta\right)R\right)} \tag{13}$$

In this equation, $R$ is a non-dimensional number that depends on the characteristics of the problem (Eq. 12), and $f$ is the relative deviation of the drag coefficient $C_d$ from $\frac{12}{Re}$ (Eq. 7). Independantly of the formulation of $f\left(Re\right)$, the relative deviation $\delta$ of $v_\infty$ from $v_\infty^{Stokes}$ is the solution of the fixed-point equation 13. Therefore, $\delta$ is a function of $R$, and only of $R$, which we will note $\delta\left(R\right)$. In other words the settling speed of particles can be expressed as:

$$v_\infty = \frac{D^2 \left(\rho_p - \rho_a\right) g}{18\mu} \times \left(1 + \delta\left(R\right)\right), \text{ with} \tag{14}$$

$$R = \frac{\rho_a D^3 \left(\rho_p - \rho_a\right) g}{36\mu^2}. \tag{15}$$

This shows that the dependance of $v_\infty$ on parameters $D$, $\rho_p$, $\rho_a$, $g$, $\mu$ and on function $f\left(Re\right)$ has a very specific form, and that, if one wants to tabulate the solutions, the values of $\delta\left(R\right)$ could be obtained once and for all solving Eq. 13 for the entire relevant range of the possible values of $R$ (where the virtual Reynolds number $R$ is defined by Eq. 12), instead of tabulating $v_\infty$ for all the possible combinations of $D$, $\rho_p$, $\rho_a$, $g$ and $\mu$. Instead of performing such a tabulation, in the next section, we will show that it is possible to find an analytical expression approximating $\delta\left(R\right)$.

## 4 The $\delta\left(R\right)$ function in the Clift-Gauvin case

Now, we proceed supposing that the expression of $C_d$ as a function of $Re$ is that of Clift and Gauvin (1971) (Eq. 5), yielding the following expression for $f\left(Re\right)$:

$$f(Re) = 0.2415 Re^{0.687} + \frac{Re}{12} \times \frac{0.42}{1 + \frac{19019}{Re^{1.16}}} \tag{16}$$

Equation 13 can then be solved iteratively, as suggested in van Boxel (1998). The resulting function $\delta\left(R\right)$ is shown on Fig. 2a. Due to the sigmoid shape of $\delta\left(R\right)$, fitting it with a logistic function of $\ln R$ is tempting, and gives relatively good results. However, a generalized logistic function gives an even better agreement with the exact solution (Fig. 2a). The equation obtained with this fit is:

$$\delta(R) \simeq -\left(1 + \mathrm{e}^{-0.4335(\ln R - 0.8921)}\right)^{-1.905} \tag{17}$$

$$\simeq -\left(1 + \left(\frac{R}{2.440}\right)^{-0.4335}\right)^{-1.905} \tag{18}$$

This expression of $\delta\left(R\right)$ yields an error relative to the exact solution $< 1\%$ up to $Re = 10$ and $< 2.5\%$ up to $Re = 10^3$ (Fig. 2b). Considering that Goossens (2019) indicate that the Clift-Gauvin empirical formulation of the drag coefficient (Eq. 5) is true within 7% of the reference drag coefficient values of Lapple and Sheperd (1940), approximating $\delta\left(R\right)$ by the generalized logistic function given in Eq. 18 is accurate enough not to degrade the evaluation of the settling speed, at least until $Re = 10^3$, which is well beyond the typical range of Reynolds number for atmospheric aerosol in free fall (Fig. 1b). Figure 2c indeed shows that the error commited by applying the fit formula (Eq. 18) instead of actually solving Eq. 13 is smaller than the discrepancy between the Clift and Gauvin (1971) formulation and the Cheng (2009) parameterization. The Clift and Gauvin (1971) and Cheng (2009) drag formulations being the two best performing formulations according to the objective scores

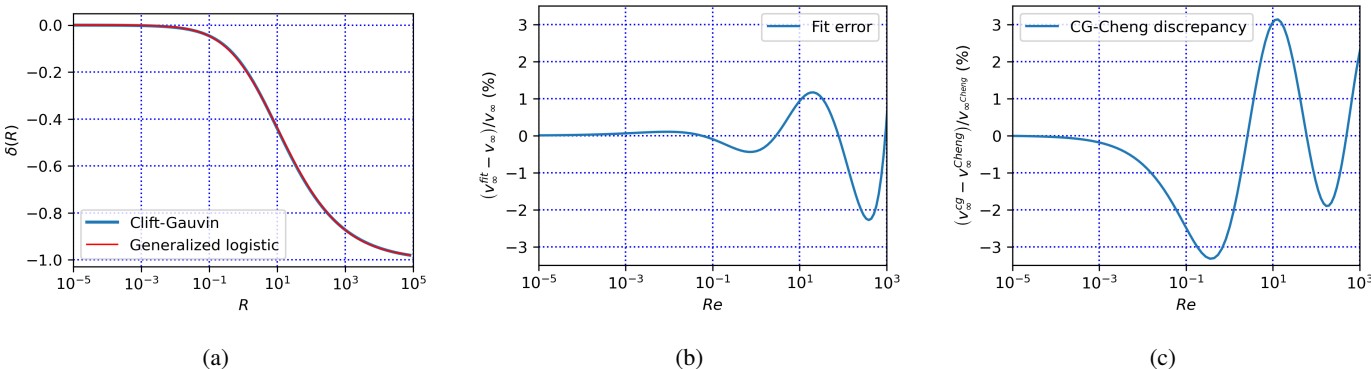

(a)              (b)              (c)

**Figure 2.** (a) $\delta(R)$ (solution of Eq. 13) as a function of $R$ when $f$ is defined from the Clift-Gauvin expression (Eq. 16). The red line is a fit of the solution by a generalized logistic function (Eq. 18). (b) percent error of the fitted expression of $v_\infty$ relative to the exact solution. (c) Percent difference of the settling speed $v_\infty$ calculated with the Clift and Gauvin (1971) parameterization relative to the one calculated with the Cheng (2009) parameterization.

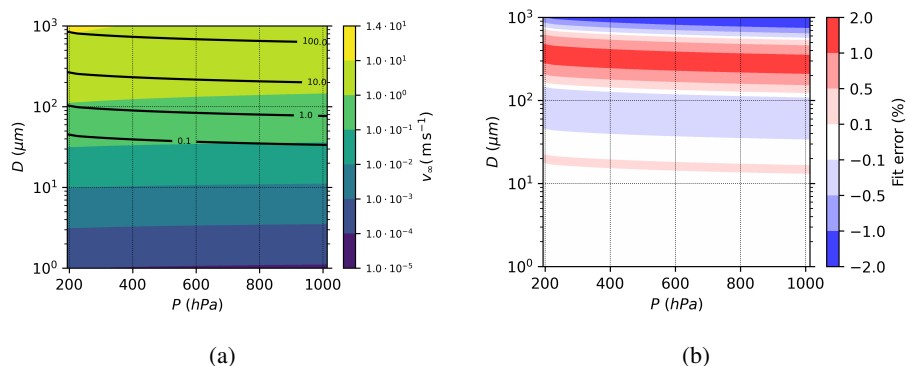

(a)                         (b)

**Figure 3.** (a) $v_\infty$ from the Clift-Gauvin expression (Eq. 16) as a function of atmospheric pressure $P$ and particle diameter $D$ for an atmospheric particle with density $\rho = 2650\,\mathrm{kg\,m^{-3}}$. The dependance of temperature on pressure has been taken from the US standard atmosphere (NOAA/NASA/USAF, 1976). Black contours are iso-$Re$ contours. (b) percent error committed by using Eq. 18 instead of solving Eq. 13.

presented in Goossens (2019) (their Table 2), this confirms that the error introduced by using Eq. 18 instead of the exact solution of Eq. 13 is smaller than the uncertainty of state-of-the-art drag coefficient formulations.

Figure 3a shows that, for all realistic particle sizes and all realistic atmospheric pressures in the troposphere, the Reynolds number is below 100, and Fig. 3b shows that the error induced by using Eq. 18 to evaluate the solution of Eq. 13 is less than 0.5% for all particles with $D < 100\,\mu\mathrm{m}$, and less than 2% for all particles with diameter less than $D < 1000\,\mu\mathrm{m}$: this shows that the domain of applicability of Eq. 18 largely covers the size range of giant dust particles (for which the typical diameter is below or around $100\,\mu\mathrm{m}$, with exceptional observations of particles with $D \simeq 400\,\mu\mathrm{m}$ as in van der Does et al. (2018)).

 **5   Inclusion of the slip correction factor**

So far, for the sake of simplicity, we have assumed that continuum fluid mechanics apply to our falling particles. As explained in, *e.g.*, Seinfeld and Pandis (1997) (Chap. 8), this assumption holds only if $Kn \ll 1$, where $Kn$ is the Knudsen number of the falling particle:

$$Kn = \frac{2\lambda}{D}, \tag{19}$$

where $\lambda$ is the mean free path of molecules in air. The mean free path in air is given by the following empirical equation as a function of pressure $P$, dynamic viscosity $\mu$ and density $\rho_a$ (Jennings, 1988):

$$\lambda = \sqrt{\frac{\pi}{8}} \times \frac{\frac{\mu}{0.4987445}}{\sqrt{P\rho_a}}. \tag{20}$$

From the Knudsen number, slip correction factors can be designed to account from non-continuous effect, turning Eq. 1 into:

$$\frac{1}{2}\frac{C_d}{C_c} A_p \rho_a v_\infty^2 = (\rho_p - \rho_a) V_p g, \tag{21}$$

where the slip correction factor $C_c$ (also known as the Cunningham (1910) correction factor) is usually estimated from the Davies (1945) expression:

$$C_c = 1 + Kn\left(1.257 + 0.4\exp\left(-\frac{1.1}{Kn}\right)\right). \tag{22}$$

From Eqs. 7 and 21, and if we introduce the Stokes terminal velocity *including the slip correction term*, $\widetilde{v}_\infty^{Stokes}$, as:

$$\widetilde{v}_\infty^{Stokes} = C_c v_\infty^{Stokes}, \tag{23}$$

we obtain:

$$v_\infty = \widetilde{v}_\infty^{Stokes} \times \frac{1}{(1 + f(Re))} \tag{24}$$

The terminal Reynolds number of the particle is equal to:

$$Re = \frac{\rho_a D \widetilde{v}_\infty^{Stokes}(1 + \delta)}{2\mu} = (1 + \delta)\frac{C_c \rho_a D^3 (\rho_p - \rho_a) g}{36\mu^2} \tag{25}$$

Let us introduce:

$$\widetilde{R} = \frac{\rho_a D \widetilde{v}_\infty^{Stokes}}{2\mu} \tag{26}$$

With these notations, $Re = (1 + \delta)\, \widetilde{R}$, and Eq. 24 becomes:

$$(1 + \delta) = \frac{1}{\left(1 + f\left((1 + \delta)\,\widetilde{R}\right)\right)} \tag{27}$$

Equation 27 is the same fixed-point equation as 13 but with parameter $\widetilde{R}$ (defined from 26) instead of $R$ (from Eq. 12). Therefore, its solution is $\delta\left(\widetilde{R}\right)$, where the $\delta$ function is the same which is represented in Fig. 2a, and can be correspondingly approximated by Eq. 18, with the same error term as represented in Fig. 2b. The fact that the introduction of the slip correction factor $C_c$ changes only very slightly the mathematical method to obtain the expression of the settling speed $v_\infty$ is a fortunate consequence of the slip correction factor being a function of the Knudsen number only, with no dependance on particle speed.

To summarize the previous development, the method we have designed to calculate $v_\infty$ including the slip correction term and the drag correction term takes the following steps:

1. Calculate $Kn$ from Eq. 19 and $C_c$ from Eq. 22

2. Calculate $\widetilde{v}_\infty^{Stokes}$ from Eq. 23 and $\widetilde{R}$ from Eq. 26

3. Calculate $v_\infty$ as $v_\infty = \widetilde{v}_\infty^{Stokes} \times \left[ 1 - \left( 1 + \left( \frac{\widetilde{R}}{2.440} \right)^{-0.4335} \right)^{-1.905} \right]$

Fig. 4 shows the impact of the slip correction term and the drag correction term on $\widetilde{v}_\infty$. Two regimes are clearly separated on this figure, with $\frac{v_\infty}{v_\infty^{Stokes}} > 1$ for the smaller particles, for which the slip correction term dominates, and the larger particles for which the drag correction term dominates. Between these two regimes exist a relatively large zone in which the departure of $v_\infty$ from $v_\infty^{Stokes}$ is less than 5%. This zone in which the Stokes equation is directly applicable covers the $3\,\mu m < D < 35\,\mu m$ diameter range at ground-level pressure, and the $10\,\mu m < D < 50\,\mu m$ range at $200\,hPa$. Some authors (*e.g.* Mallios et al. (2020)) argue that the slip correction factor $C_c$ should be applied only in the Stokes regime ($Re < 0.1$). However, in Fig. 4, it has been applied regardless of the Reynolds number. Analysis of Fig. 4 shows that the choice of applying or not the slip correction factor for $Re > 0.1$ has little consequence at least in the troposphere, because in all the portion of the pressure-diameter diagram where $Re > 0.1$ (above the red line in Fig. 4), $C_c$ is comprised between 1 and 1.02, so that applying or not this factor which is extremely close to 1 has no practical consequence on the simulated value of $v_\infty$.

## 6 Implementation and computational efficiency

Apart from its simplicity, another possible advantage of the straightfoward evaluation of $v_\infty$ using the three above-defined steps is its computational efficiency compared to the currently available methods:

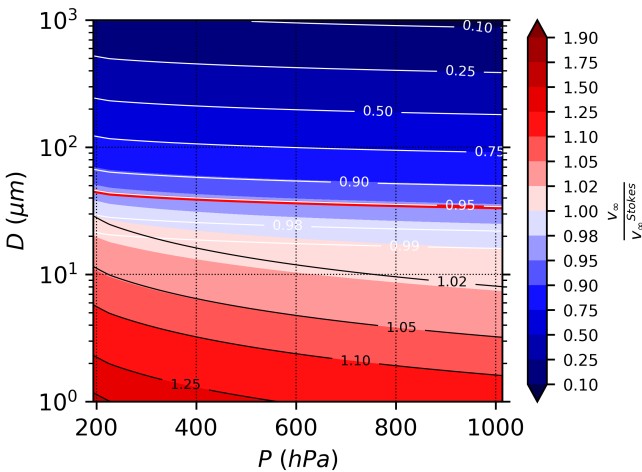

**Figure 4.** Ratio $\frac{v_\infty}{v_\infty^{Stokes}}$ as a function of atmospheric pressure and particle diameter for a spherical particle with $\rho_p = 2650\,\mathrm{kg\,m^{-3}}$ (color shades). Black contours are iso-lines for the slip correction factor $C_c = \frac{\widetilde{v}_\infty^{Stokes}}{v_\infty^{Stokes}}$, white lines are iso-lines of $1+\delta\left(\widetilde{R}\right) = \frac{v_\infty}{\widetilde{v}_\infty^{Stokes}}$. The red line corresponds to $Re = 0.1$, with $Re > 0.1$ above this line and $Re < 0.1$ below.

- *Bisection method*, finding $Re$ as the solution to $Re = \frac{\widetilde{R}}{1+f(Re)}$ starting from $Re \in \left[0; \widetilde{R}\right]$ and successively cutting the solution interval $[Re_\mathrm{m}; Re_\mathrm{M}]$ into halves until the relative error on speed becomes smaller than $\varepsilon$: $\frac{Re_\mathrm{M} - Re_\mathrm{m}}{Re_\mathrm{m}} < \varepsilon$; then $v_\infty = \widetilde{v}_\infty^{Stokes}\frac{Re}{\widetilde{R}}$

- *Fixed-point method*, finding the solution $\delta$ to Eq. 27 starting from $\delta_0 = 0$ and iterating with $\delta_{i+1} = \frac{1}{1+f\left(\widetilde{R}\times(1+\delta_i)\right)} - 1$, until the relative error on speed becomes smaller than $\varepsilon$: $\frac{|\delta_{i+1} - \delta_i|}{1+\delta_{i+1}} < \varepsilon$. Then, $v_\infty = \widetilde{v}_\infty^{Stokes}(1+\delta)$;

- *AerSett method*, calculating directly $v_\infty = \widetilde{v}_\infty^{Stokes} \times \left[1 - \left(1 + \left(\frac{\widetilde{R}}{2.440}\right)^{-0.4335}\right)^{-1.905}\right]$.

The bisection and fixed-point methods depend on a tolerance parameter $\varepsilon$, which we have set to $\varepsilon = 0.02$, corresponding to the maximal error of 2% observed using the AerSett method (Fig. 3b).

A Fortran code has been designed to estimate the calculation cost for these three methods (Fig. 5). This has been done by performing the calculation of $v_\infty$ for $10^8$ random values of $D$, at a random altitude $z$ in the atmosphere between 0 and 12000 m. The pressure and temperature values $P(z)$ and $T(z)$ are estimated from the US Standard Atmospheric profile (NOAA/NASA/USAF, 1976).

To optimize calculation speed, we have observed that for $\widetilde{R} < 0.0116$ we have $1+\delta\left(\widetilde{R}\right) > 0.99$, so that for all three methods we speed up the calculation by assuming that $v_\infty = \widetilde{v}_\infty^{Stokes}$ when $\widetilde{R} < 0.0116$. As visible on Fig. 4 (see the 0.99 iso-line of $1+\delta\left(\widetilde{R}\right)$), this means that for all particles with a diameter $D < 20\,\mu m$ (the precise threshold value depends on pressure and temperature), no drag correction factor is applied.

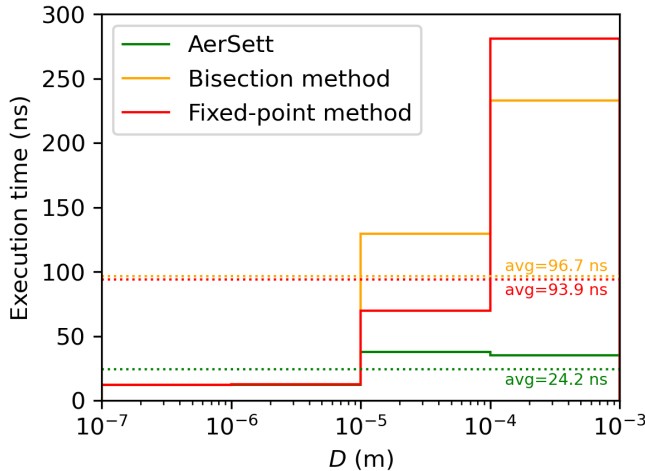

**Figure 5.** Execution time in ns for the three numerical methods described in Section 6 as a function of $D$. The estimate has been performed for 4 ranges of $D$ (0.1-1 $\mu$m; 1-10 $\mu$m; 10-100 $\mu$m; 100-1000 $\mu$m). In each of these intervals, the calculation of $v_\infty$ as a function of $D$, $\rho_p$, $\rho_a$, $\lambda$ and $\mu$ has been performed $10^8$ times with each method, for $10^8$ values of $D$ selected randomly within the range. The execution time in ns corresponds to the total execution time (for $10^8$ calls to the function) divided by the number of calls ($10^8$) so that it represents an estimate of the execution time for one call to the function. The test has been performed on a laptop with an Intel Core i7-1165G7 CPU.

Fig. 5 showing that, for $D < 10\,\mu$m the computation time (around 12 ns per call) is the same for the three methods, which is expected since the drag-correction factor is not calculated for these diameters, which result in $\widetilde{R} < 0.0116$. In average over the entire interval, the bisection method and the fixed-point method are about 4 times slower than the AerSett method. For $D > 100\,\mu$m, the bisection method is 6.6 times slower than AerSett, and the fixed-point method is 8 times slower. It is also worth noting on Fig. 5 that the fixed-point method is faster than the bisection method for small values of $D$, but slower for large values of D. The average number of iterations to obtain convergence increases from 4.5 ($10\,\mu$m $< D < 100\,\mu$m) to 7.8 ($100\,\mu$m $< D < 1000\,\mu$m) for the bisection method, and from 1.5 ($10\,\mu$m $< D < 100\,\mu$m) to 7.4 ($100\,\mu$m $< D < 1000\,\mu$m) for the bisection method. This sharper increase in the number of iterations to obtain convergence in the fixed-point method explains why bisection becomes more efficient for large values of $D$.

## 7 Conclusions

As a conclusion, we have found that the following method is suitable to evaluate the settling speed of spherical aerosol particles in the atmosphere, $v_\infty$:

$$\widetilde{v}_\infty^{Stokes} = C_c \frac{D^2 (\rho_p - \rho_a) g}{18\mu} \; ; \tag{28}$$

$$\widetilde{R} = \frac{\rho_a D \widetilde{v}_\infty^{Stokes}}{2\mu} \; ; \tag{29}$$

$$v_\infty = \widetilde{v}_\infty^{Stokes} \times \left[ 1 - \left( 1 + \left( \frac{\widetilde{R}}{2.440} \right)^{-0.4335} \right)^{-1.905} \right] . \tag{30}$$

The slip correction factor $C_c$ can be obtained by the Davies (1945) formula (Eq. 22). To reduce computation time, for $\widetilde{R} < 0.116$, Eq. 30 can be replaced by just $v_\infty = \widetilde{v}_\infty^{Stokes}$, changing the result by less than $1\%$.

Equations 28, 29 and 30 constitute the AerSett model v1.0 (AerSett for AERosol SETTling). The error induced by applying

this model compared to an iterative calculation of $v_\infty$ is less than $0.5\%$ for particles with diameter $D < 100\,\mu m$ and less than $2\%$ for particles with $D < 1000\,\mu m$ (Fig. 3b). Particles with larger diameters fall so rapidly that they are not relevant as atmospheric aerosol: other parameterizations exist for the falling hydrometeors (Khvorostyanov and Curry (2005)), taking into account the shape of snow flakes, the deformation of raindrops due to their speed etc. We have shown that the error due to using Eq. 30 is smaller than the uncertainty that exists between different state-of-the-art formulations of the drag coefficient,

showing that this error is not a problem for modelling.

Eq. 30 takes into account both the slip correction factor (for small particles) and the large-particle drag correction factor. Fig. 4 shows that for particles smaller than $D \simeq 10\,\mu m$ the slip correction factor dominates, while for particles larger than $20\,\mu m$ the large-particle drag correction factor dominates. The reduction in settling speed relative to $v_\infty^{Stokes}$ reaches about $25\%$ for a particle with diameter $D \simeq 100\,\mu m$, a typical diameter for giant dust particles. So, if chemistry-transport models

are to represent the observed giant dust particles (which is still a challenge), large-particle drag correction for the gravitational settling speed needs to be taken into account, and we think that Eq. 30, valid for all spherical particles with $D < 1000\,\mu m$ and at least from the surface to $p = 200\,\text{hPa}$ (Figs. 3b and 4), is the simplest available method to do it. While designed specifically to take into account large-particle drag correction, Eq. 30 includes the slip correction term as well, which is critical for the finer atmospheric aerosol with $D < 10\,\mu m$. Using Eq. 30 yields an error smaller than $2\%$ relative to the exact combined effect

of the Davies (1945) slip correction factor and the Clift and Gauvin (1971) large-particle drag correction factor. Therefore it is possible to use this formulation systematically for spherical particles in chemistry-transport models. An implementation of Equations 28-30 in a Fortran module along with the necessary thermodynamical calculations is available for download and use (see the *code availability* section). We have shown (Section 6, and in particular Fig. 5) that the use of this Fortran routine permits to gain a factor 4 (in general) to 8 (for particles with $D > 100\,\mu m$) relative to bisection or fixed-point methods.

To go further in understanding the settling speed of giant dust particles and be able to represent them in chemistry-transport models, it is needed to extend simple models such as AerSett to the case of non-spherical particles, and give simple and

straightforward estimates of the drag correction factor $\delta$ not only as a function of particle density and diameter, as in the present study, but also including other factors such as particle eccentricity which have been shown to have a strong impact on the settling speed of dust particles (Mallios et al. (2020) and references therein). To solve the persistent mystery of the

265 processes allowing giant dust particles to stay airborne over long distances, new findings on physical processes such as the electric charges of the particles and their effect on settling velocities are still needed.

*Code availability.* All the simulations and figures have been performed with python scripts and Fortran code available according to the GNU General Public License v3.0 at the following doi: 10.5281/zenodo.7535171

All the available versions of the AerSett Fortran module are available according to the GNU General Public License v3.0 at the following

doi: 10.5281/zenodo.7535114

*Author contributions.* All authors have contributed to the article by stirring the ideas, writing and correcting the paper and providing bibliographical reference. SM has developed the Python scripts used to produce the plots.

*Competing interests.* None.

*Acknowledgements.* The Authors acknowledge the developers of the python modules `thermo` (Caleb Bell and Contributors[1] and `scipy` (Virtanen et al. (2020)), used for our numerical treatment of the problem.

---

[1] https://github.com/CalebBell/thermo

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
