# Peer review of "AerSett v1.0: A simple and straightforward model for the settling speed of big spherical atmospheric aerosol."

_Geoscientific Model Development, 2022_

## Referee Comment (RC1)

**General Comments**

Review of the paper " AerSett v1.0: A simple and straightforward model for the settling speed of big spherical atmospheric aerosol. " by Mailler et al.

In this manuscript the authors derive a mathematical expression for the calculation of the settling velocity of spherical atmospheric aerosols up to diameter of 1mm. They showed that the derived direct formula gives results within 2% of the exact solution obtained by numerical methods, and can be used in global transport models to improve their computational speed.

The paper is well motivated, and in principle logically well organized. There are some parts that need to be expanded and rewritten, such as the Introduction and the Conclusion sections (please see my comments below). The abstract summarizes the presented results, the bibliography corresponds to the good quality of the paper (although it needs to be enriched). The figures are with good quality of precise description and illustrate well the quality of the paper and bring the necessary information.

I suggest the publication of this work after some clarifications and additions are made.

**Specific comments**

- Section 1.
    - Introduction section needs to be better organized and rewritten. It does not place the study in a broader scientific context, and it does not highlight the significance of this work.

    - Line 10: The authors state that the goal of this article is the study of the terminal velocity of spherical aerosol particles. Why? Why do we care? Why is this important?

    - Line 16: References are missing. Moreover, definitions of "small dust particles" and "giant dust particles" are missing.

    - Lines 18-21. This sentence does not make sense. The authors list some reasons (without proper reference and a small description) that contribute to an underestimation. Underestimation of what? More over they mention a recent focus on giant particles without providing some references to show this focus. Finally, they state that this focus on giant particles highlights the need of a robust and efficient way of calculating the settling speed of large particles. This is not quite accurate. As Adebiyi and Kok (2020) show, the large particles are not represented at all in global models. This means that the top priority is the need to include the effects of these particles to the models, with the calculation of their terminal velocity being just one of the aspects that need to be addressed. Of course a robust and efficient way of this calculation is needed, but right now there is no way at all.

- o Lines 25-28. The authors discuss the work by Drakaki et al. (2022). They say that in the referenced work the drag coefficient by Clift and Gauvin (1971) has been used by a bisection method. Then they say that Drakaki et al. (2022) highlight the importance of including large particles (for what?), and then they say again that the Clift and Gauvin (1971) drag coefficient has been used. These lines need to be rewritten.

  - o Lines 28-31. In the previous lines the authors state that Drakaki et al. (2022) showed the importance of including large particles, but here they state that Drakaki et al. (2022) remark that much better agreement between model and observations is reached when, apart for applying the Clift and Gauvin (1971) drag correction factor, the settling speed of dust particles is reduced by an empirical factor of 80% . 80% is a large artificial reduction. So, how important is the change of the drag coefficient? I find the way that the work by Drakaki et al. (2022) is presented by the authors to be confusing and not accurate.

- Section 2.
  - o Lines 41-42. Mallios et al. (2020) as well as Drakaki et al. (2022) include the slip correction factor in the drag equation, because it is crucial for particles with Reynolds numbers less than 1. Why do the authors have omitted this factor?

  - o Line 61. van Boxel (1998) (the reference is not correctly presented in this line) does not describe the iterative method. They  just mention that an iterative method has been used to solve the non linear equations. So, what iterative method can be used to solve Equation 1  (and not 2 as mentioned in the text)?

  - o Lines 62-63. What deviation is considered strong by the authors? According to Figure 1c there is 10% deviation from Stokes' solution in the case of particles with D around 40 μm, and 20% in the case of particles with D around 80 μm. Especially the 10% deviation is not strong and is similar to the accuracy of the Clift and Gauvin drag coefficient expression.

  - o Lines 68-70. The authors need to give details on the iteration method that has been used. They also need to be more precise on the number of iterations that are needed for the solution of Eq. 1. There are several root finding algorithms that can be extremely robust (e.g. Brent's algorithm) or extremely slow (e.g. bisection method).

- Section 4.
  - o Line 104. The iterative method should be properly presented and referenced, because the presented reference of van Boxel (1998) does not describe this method.

- Section 5.

o The authors need to present more clearly the significance of this work. The derived mathematical expression is valid only in the case of spherical particles, without taking account the slip correction factor for low Reynolds numbers, and by considering just the gravity and the drag forces. This means that it cannot be generalized to other shapes, and it is not valid when other forces (e.g. the electrical forces) act on the particle. So, what is the benefit of using this expression of limited applicability? What is the computational time gain against robust iterative numerical methods that can solve more general problems?

o Line 144. The slip correction factor should be properly introduced and described.

o Line 145. The Knudsen number should be properly introduced, defined and referenced.

**Technical corrections**

- Line 10. computanionnally -> computationally.

- Line 61. Proper reference of van Boxel is missing.

- Line 86. "...the particle if it would have if it would be falling..." -> "...the particle it would have if it would be falling...".

- Line 150. excentricity -> eccentricity.

---

## Referee Comment (RC2)

**General comments**

Review of the paper "AerSett v1.0: A simple and straightforward model for the settling speed of big spherical atmospheric aerosol " by Mailler et al.

In this scientific paper authors present an alternative methodology for the calculation of the terminal velocities of spherical particles beyond the Stokes regime. The method includes a mathematical expression that approximates the non-analytical solution of the 1-D (vertical) equation of motion by circumventing the utilization of an iteration method which is originally needed. They show that the error of the methodology is acceptable with its maximum value to be around 2% for particles up to 1000µm.

The paper is well-written, well-organized, with a straightforward abstract and fulfills the main goal of the article. Although, some adjustments should be done in the Introduction section. The results are clearly presented in good-quality graphs and the conclusions are well established.

I suggest the publication of this work after some minor revisions.

**Specific comments**

Introduction in general: I suggest the authors to change the order of the first two paragraphs. This will help the reader to understand the topic's background and prepare him for more detailed and specific information that is given later. Also the introduction should include more papers of prior research on large dust particles.

Lines 10-11: What is large-particle correction? Authors should introduce that term in order a less engaged reader can understand better the meaning of the sentence.

Line 16: Please give the definition of giant particles.

Lines 16-17: References are needed here.

Lines 18-19: Why do we care about the missing from the models coarse dust particles. How do they affect the physical processes in the atmosphere?

Line 26: The authors state that Drakaki et al. (2022) use Clift and Gauvin (1971) correction and performed the bisection method once for each model size bin. The word "once" is a little confusing, since viscosity depends on pressure and temperature, which changes at each time step and in each model grid box. Thus, the terminal velocity is calculated accordingly at each time step, in each model grid box and for each model size bin, adapting the bisection method. This makes the code even more time consuming.

Line 51: The expression for air viscosity is missing.

Line 60: For Re<0.1 the consideration of free-slip correction should be added as it is described in Drakaki et al. (2022) and Mallios et al. (2020). Why did you omit it in both Stokes and Clift and Gauvin expressions? Could the consideration of the free-slip correction possibly change the methodology? By not including the slip-free correction, makes the methodology valid only for Re ≥ 0.1.

Line 61: Please describe in detail the iterative method you used. Also in line 68.

Line 141: Please define the exact ranges that the expression is valid.

Line 142: Can you provide an estimation of the computational benefit of the method?

**Technical corrections**

Line 61: Eq.2 instead of Eq.1.

Line 61: Please insert the citation o Van Boxel properly.

Line 150: "excentricity" to "eccentricity"

---

## Author Comment (AC1)

**Answer to RC1 comment on the manuscript "**AerSett v1.0: A simple and straightforward model for the settling speed of big spherical atmospheric aerosol.**"**

**Jan. 13, 2023**

We are grateful to Anonymous Reviewer #1 for his/her careful Review of our manuscript.

**1 General comments**

In this manuscript the authors derive a mathematical expression for the calculation of the settling velocity of spherical atmospheric aerosols up to diameter of 1mm. They showed that the derived direct formula gives results within 2% of the exact solution obtained by numerical methods, and can be used in global transport models to improve their computational speed. The paper is well motivated, and in principle logically well organized. There are some parts that need to be expanded and rewritten, such as the Introduction and the Conclusion sections (please see my comments below). The abstract summarizes the presented results, the bibliography corresponds to the good quality of the paper (although it needs to be enriched). The figures are with good quality of precise description and illustrate well the quality of the paper and bring the necessary information. I suggest the publication of this work after some clarifications and additions are made.

We are grateful to both Reviewers for their positive appreciation of our manuscript and for their comments, helping us propose an improved version of the manuscript. In particular, we have added two new sections to address the shortcomings identified by the both Reviewers in the discussion:

**Section 5: Inclusion of the slip-correction factor**

This section addresses one of the the main comments of both Reviewers (Reviewer 1: "Lines 41-42. Mallios et al. (2020) as well as Drakaki et al. (2022) include the slip correction factor in the drag equation, because it is crucial for particles with Reynolds numbers less than 1. Why do the authors have omitted this factor?", Reviewer 2, "Line 60: For Re¡0.1 the consideration of free-slip correction should be added as it is described in Drakaki et al. (2022) and Mallios et al. (2020). Why did you omit it in both Stokes and Clift and Gauvin expressions? Could the consideration of the free-slip correction possibly change the methodology?

By not including the slip-free correction, makes the methodology valid only for $Re \geq 0.1$.”), which as they correctly assert would be a critical limitation of our method. Therefore, we present a modified version of our method to explicitly include this effect.

**Section 6: Implementation and computational efficiency**

This new section also addresses a request by both Reviewers (Reviewer 1: “ What is the computational time gain against robust iterative numerical methods that can solve more general problems?”, Reviewer 2, “Can you provide an estimation of the computational benefit of the method?”

We feel that with these additions, and rewriting the introduction that was found confusing by both Reviewers, this paper has improved considerably and now provides an out-of-the-box solution to modellers needing to estimate correctly the settling speed of spherical aerosol for the entire range of atmospheric conditions and relevant diameters for atmospheric aerosols.

Also, please note that there was a problem with the transcription of our numerical method in the paper: in the formulae describing our numerical method, a coefficient was mistakenly written as 0.4335 instead of $-0.4335$. There was no problem in our numerical calculations or in the corresponding scripts and figures, only on their transcription into litteral formulae in the paper. This has been fixed in this new version.

**2 specific comments**

**2.1 Section 1**

Introduction section needs to be better organized and rewritten. It does not place the study in a broader scientific context, and it does not highlight the significance of this work.

, The outline of the introduction was indeed confusing, navigating back and forth between different topics. We used the detailed suggestions by Reviewers 1 and 2 to reorganize and add more substance to the Introduction where needed. The introduction is now organized following a clearer outline.

First, we explain why it is important to calculate accurately the settling velocity of aerosols (as it governs dry deposition, which is their main sinks)

Then we briefly introduce some of the bibliography on giant dust particles, without the pretention to give a bibliographic overview of this topic (which as precised in the revised version is beyond the scope of our sudy: the reader is referred to other studies for more bibliography). The point of this paragraph is to justify that:

1. Giant dust particles exist in the atsmosphere

2. they are not only a curiosity, but they also play a geophysical role

After that, we put another paragraph explaining why modelling giant dust particles requires to take intro account large-particle correction factor to the Stokes law, and how this has been done so far.

When this is done, we present the goal of the paper, and its outline.

Line 10: The authors state that the goal of this article is the study of the terminal velocity of spherical aerosol particles. Why? Why do we care? Why is this important?

We have tried to rewrite the Introduction in a more logical and straightforward fashion, following the logical steps described above. Specifically, the first paragraph of the revised introduction now explains why modelling the settling velocity of aerosol accurately is crucial:

"One of the main purposes of the chemistry-transport modeling is to simulate as accurately as possible the aerosol concentration in the atmosphere. The settling velocity of aerosol is a key driver of their dry removal from the atmosphere ([Zhang et al., 2001]). Dry removal being the only sink for atmospheric aerosol under dry conditions, any error on representing the dry deposition velocity of atmospheric aerosol will have a direct impact on modelled concentrations."

Line 16: References are missing. Moreover, definitions of "small dust particles" and "giant dust particles" are missing.

We have added the definitions of accumulation, coarse and giant dust following [Ryder et al., 2019] in the introduction. We for sure did not include all the relevant bibliography for giant dust: giant dust is not the goal of the present article, but the argument we propose in the introduction to explain that it is important to be able to calculate the settling speed for such particles quickly and accurately. in the revised version, we explicitly redirect the reader to [Ryder et al., 2019], [van der Does et al., 2018] and [Drakaki et al., 2022] for further references.

Lines 18-21. This sentence does not make sense. The authors list some reasons (without proper reference and a small description) that contribute to an underestimation. Underestimation of what? More over they mention a recent focus on giant particles without providing some references to show this focus. Finally, they state that this focus on giant particles highlights the need of a robust and efficient way of calculating the settling speed of large particles. This is not quite accurate. As Adebiyi and Kok (2020) show, the large particles are not represented at all in global models. This means that the top priority is the need to include the effects of these particles to the models, with the calculation of their terminal velocity being just one of the aspects that need to be addressed. Of course a robust and efficient way of this calculation is needed, but right now there is no way at all.

We agree that the logical step here was confusing here. With the general rewriting of the introduction, the sentence referred to has disappeared, some factors hindering so far the representation of giant dust particles in models is presented.

We do not claim that the need of having a robust and efficient way to calculate the settling speed is the most urgent or the key blocking problem, it is of course just one of the aspects, as it is now clear in the revised version of the introduction:

Since the important impact of giant dust particles on the dust concentration and optical effect has been demonstrated (e.g. [Ryder et al., 2019]), there is an emerging need to solve the problems that hinder the representation of giant dust particles in CTMs and General circulation models. Designing a robust and efficient method to calculate the settling speed of giant dust particles is a step in this direction.

Lines 25-28. The authors discuss the work by Drakaki et al. (2022). They say that in the referenced work the drag coefficient by Clift and Gauvin (1971) has been used by a bisection method. Then they say that Drakaki et al. (2022) highlight the importance of including large particles (for what?), and then they say again that the Clift and Gauvin (1971) drag coefficient has been used. These lines need to be rewritten.

On several occasions in the introduction, the manuscript was going back and forth this way. This is why we have totally reorganized the introduction, we hope it is more clearly structured now.

Lines 28-31. In the previous lines the authors state that Drakaki et al. (2022) showed the importance of including large particles, but here they state that Drakaki et al. (2022) remark that much better agreement between model and observations is reached when, apart for applying the Clift and Gauvin (1971) drag correction factor, the settling speed of dust particles is reduced by an empirical factor of 80% . 80% is a large artificial reduction. So, how important is the change of the drag coefficient? I find the way that the work by Drakaki et al. (2022) is presented by the authors to be confusing and not accurate.

In order to better convey the conclusions of [Drakaki et al., 2022], we now use a wording very close to the wording of these authors to decribe their conclusions:

"For example, Drakaki et al. (2022) show that the WRFV4.2.1 model with a version of the GOCART-AFWA dust scheme modified to include coarse and giant dust particles underestimates the lifetime of coarse and giant dust particles in the atmosphere. They show that their simulation results 30 are closer to observation when they include an artificial reduction of particles' settling velocities by 60% to 80% (depending on the diameter). This reduction is a way to account for underrepresented mechanisms such as non-sphericity of particles (Mallios et al., 2020), or their electric charges, which have been discussed as possible factors explaining a longer atmospheric lifetime of coarse dust particles (Adebiyi and Kok, 2020)."

We hope that the following formulation conveys the finding of Drakaki et al. (2022) better.

**2.2   Section 2**

Lines 41-42. Mallios et al. (2020) as well as Drakaki et al. (2022) include the slip correction factor in the drag equation, because it is crucial for particles with Reynolds numbers less than 1. Why do the authors have omitted this factor?

As mentioned in, e.g., [Drakaki et al., 2022] and [Mallios et al., 2020], the slip-correction factor should in principle be applied only to the Stokes regime ($Re < 0.1$). On the contrary, the drag-correction factors (similar to [Clift and Gauvin, 1971]) describe the transition of the flow around the falling particle from the laminar to the turbulent regime ([Goossens, 2019]). Therefore, it is not physically shocking to study one effect independantly of the other.

However, we agree with the Reviewer that rather than leaving the user with choices of applying no correction, free-slip correction, large-particle correction or both, it is better to provide a method that permits to include seamlessly both correction terms and could be used for all atmospheric particles.

For this reason, we have included a new section in the article (Section 5 of the revised manuscript) extending our findings with the inclusion of the Cunningham free slip correction term so that the method we provide

becomes applicable to all (spherical) atmospheric aerosol (Fig. 4 of the revised manuscript).

We are grateful for the Reviewer to have pointed this improvement direction, because we think including it will greatly improve the usability of the method we propose.

Line 61. van Boxel (1998) (the reference is not correctly presented in this line) does not describe the iterative method. They just mention that an iterative method has been used to solve the non linear equations. So, what iterative method can be used to solve Equation 1 (and not 2 as mentioned in the text)?

The Reviewer is correct, we had the false memory of getting this iterative method from [van Boxel, 1998] but actually this author just gives the hint that an iterative method can be used, but does not describe the method. In Section 6 of the revised manuscript, we explained how we have implemented this fixed-point method, as well as the bisection method, and we examine the computation time of these different methods.

Lines 62-63. What deviation is considered strong by the authors? According to Figure 1c there is 10% deviation from Stokes' solution in the case of particles with D around 40 $\mu$m, and 20% in the case of particles with D around 80 $\mu$m. Especially the 10% deviation is not strong and is similar to the accuracy of the Clift and Gauvin drag coefficient expression.

The Reviewer correctly points that "strong" in this context is not adapted (the difference is less than 1% at this point). we have rephrased the sentence in a more objective way:

"Fig. 2a shows that the Stokes equation (Eq. 6) gives excellent results for $D < 20\,\mu$m, but that deviations from it due to the departure of the drag coefficient from Eq. 2 gradually arise when $D$ exceeds $20\,\mu$m, reaching $-10\%$ when $D \simeq 50\,\mu$m, $-30\%$ when $D \simeq 100\,\mu$m, and $-90\%$ when $D \simeq 1000\,\mu$m (Fig. 2c)."

Lines 68-70. The authors need to give details on the iteration method that has been used. They also need to be more precise on the number of iterations that are needed for the solution of Eq. 1. There are several root finding algorithms that can be extremely robust (e.g. Brent's algorithm) or extremely slow (e.g. bisection method).

These details (definition of the two methods we have tested apart of our own method) are now given in the new Section 6 about the computational performance of the method. Elements on the number of iterations needed for the convergence of the fixed-point iterative method are also given in that section. We have not investigated more root-finding algorithms than bisection and fixed-point iterations.

**2.3 Section 4**

Line 104. The iterative method should be properly presented and referenced, because the presented reference of van Boxel (1998) does not describe this method.

These precisions are now provided in Section 6.

**2.4 Conclusion**

The authors need to present more clearly the significance of this work. The derived mathematical expression is valid only in the case of spherical particles, without taking account the slip correction factor for low Reynolds numbers, and by considering just the gravity and the drag forces. This means that it cannot be generalized to other shapes, and it is not valid when other forces (e.g. the electrical forces) act on the particle. So, what is the benefit of using this expression of limited applicability? What is the computational time gain against robust iterative numerical methods that can solve more general problems?

> The main limitation raised by the Reviewer has been lifted in the new version of the manuscript, by explicitly including the slip-correction factor in the new version of the method that is summarized in the Conclusion. We mention generalization to other shapes (including at least an excentricity parameter) as the main development direction for this method, since [Mallios et al., 2020] shows that much information has been obtained on this question, which may permit to design a closed expression for oblate/prolate particles as well. We therefore do not feel that the present method "cannot be generalized to other shapes".
>
> Regarding electric forces, we feel that this is still unfortunately a question for process studies, quantifying the charges that could be carried by dust particles and their consequences on their settling velocity. Since no definitive observational results are available yet on this question, we feel that the scientific conditions required before proposing a mathematical formulation including these electrical effect are not met for the moment. We have added a sentence about including electric forces as well in the conclusion:
>
> "To solve the persistent mystery of the processes allowing giant dust particles to stay airborne over long distances, new findings on physical processes such as the electric charges of the particles and ther effect on settling velocities are still needed."

Line 144. The slip correction factor should be properly introduced and described.

Line 145. The Knudsen number should be properly introduced, defined and referenced.

> This is the object of Section 5 in the revised manuscript where the Knudsen number and the slip correction factor are introduced defined, referenced

Best regards,

The Authors.

**References**

[Clift and Gauvin, 1971] Clift, R. and Gauvin, W. H. (1971). Motion of entrained particles in gas streams. *The Canadian Journal of Chemical Engineering*, 49.

[Drakaki et al., 2022] Drakaki, E., Amiridis, V., Tsekeri, A., Gkikas, A., Proestakis, E., Mallios, S., Solomos, S., Spyrou, C., Marinou, E., Ryder, C. L., Bouris, D., and Katsafados, P. (2022). Modeling coarse and giant desert dust particles. *Atmospheric Chemistry and Physics*, 22(18):12727–12748.

[Goossens, 2019] Goossens, W. R. (2019). Review of the empirical correlations for the drag coefficient of rigid spheres. *Powder Technology*, 352:350–359.

[Mallios et al., 2020] Mallios, S. A., Drakaki, E., and Amiridis, V. (2020). Effects of dust particle sphericity and orientation on their gravitational settling in the earth's atmosphere. *Journal of Aerosol Science*, 150:105634.

[Ryder et al., 2019] Ryder, C. L., Highwood, E. J., Walser, A., Seibert, P., Philipp, A., and Weinzierl, B. (2019). Coarse and giant particles are ubiquitous in saharan dust export regions and are radiatively significant over the sahara. *Atmospheric Chemistry and Physics*, 19(24):15353–15376.

[van Boxel, 1998] van Boxel, J. (1998). Numerical model for the fall speed of raindrops in a rainfall simulator. Technical Report 1998/1, I. C. E. special report.

[van der Does et al., 2018] van der Does, M., Knippertz, P., Zschenderlein, P., Giles Harrison, R., and Stuut, J.-B. W. (2018). The mysterious long-range transport of giant mineral dust particles. *Science Advances*.

[Zhang et al., 2001] Zhang, L., Gong, S., Padro, J., and Barrie, L. (2001). A size-segregated particle dry deposition scheme for an atmospheric aerosol module. *Atmospheric Environemnt*, 35(3):549–560.

---

## Author Comment (AC2)

**Answer to RC2 comment on the manuscript "**AerSett v1.0: A simple and straightforward model for the settling speed of big spherical atmospheric aerosol.**"**

**Jan. 13, 2023**

We are grateful to Anonymous Reviewer #2 for his/her careful Review of our manuscript.

**1 General comments**

In this scientific paper authors present an alternative methodology for the calculation of the terminal velocities of spherical particles beyond the Stokes regime. The method includes a mathematical expression that approximates the non-analytical solution of the 1-D (vertical) equation of motion by circumventing the utilization of an iteration method which is originally needed. They show that the error of the methodology is acceptable with its maximum value to be around 2% for particles up to 1000 $\mu$m. The paper is well-written, well-organized, with a straightforward abstract and fulfills the main goal of the article. Although, some adjustments should be done in the Introduction section. The results are clearly presented in good-quality graphs and the conclusions are well established.

I suggest the publication of this work after some minor revisions.

> We are grateful to both Reviewers for their positive appreciation of our manuscript and for their comments, helping us propose a much improved version of the manuscript. In particular, we have added two new sections to address the shortcomings identified by the both Reviewers in the discussion:
>
> **Section 5: Inclusion of the slip-correction factor**
>
> This section addresses one of the the main comments of both Reviewers (Reviewer 1: "Lines 41-42. Mallios et al. (2020) as well as Drakaki et al. (2022) include the slip correction factor in the drag equation, because it is crucial for particles with Reynolds numbers less than 1. Why do the authors have omitted this factor?", Reviewer 2, "Line 60: For Re¡0.1 the consideration of free-slip correction should be added as it is described in Drakaki et al. (2022) and Mallios et al. (2020). Why did you omit it in both Stokes and Clift and Gauvin expressions? Could the consideration of the free-slip correction possibly change the methodology? By not including the slip-free correction, makes the methodology valid only for $Re \geq 0.1$."), which as they correctly assert would

be a critical limitation of our method. Therefore, we present a modified version of our method to explicitly include this effect.

**Section 6: Implementation and computational efficiency**

This new section also addresses a request by both Reviewers (Reviewer 1: " What is the computational time gain against robust iterative numerical methods that can solve more general problems?", Reviewer 2, "Can you provide an estimation of the computational benefit of the method?"

We feel that with these additions, and rewriting the introduction that was found confusing by both Reviewers, this paper has improved considerably and now provides an out-of-the-box solution to modellers needing to estimate correctly the settling speed of spherical aerosol for the entire range of atmospheric conditions and relevant diameters for atmospheric aerosols.

Also, please note that there was a problem with the transcription of our numerical method in the paper: in the formulae describing our numerical method, a coefficient was mistakenly written as $0.4335$ instead of $-0.4335$. There was no problem in our numerical calculations or in the corresponding scripts and figures, only on their transcription into litteral formulae in the paper. This has been fixed in this new version.

**2 specific comments**

**2.1 Section 1**

Introduction in general: I suggest the authors to change the order of the first two paragraphs. This will help the reader to understand the topic's background and prepare him for more detailed and specific information that is given later. Also the introduction should include more papers of prior research on large dust particles.

> As also noted by Reviewer 1, the outline of the introduction was indeed confusing, navigating back and forth between different topics. We used the detailed suggestions by Reviewers 1 and 2 to reorganize and add more substance to the Introduction where needed. the introduction is now organized following a clearer outline.
>
> First, we explain why it is important to calculate accurately the settling velocity of aerosols (as it governs dry deposition, which is their main sinks)
>
> Then we briefly introduce some of the bibliography on giant dust particles, without the pretention to give a bibliographic overview of this topic (which as precised in the revised version is beyond the scope of our sudy: the reader is referred to other studies for more bibliography). The point of this paragraph is to justify that:

1. Giant dust particles exist in the atsmosphere

2. they are not only a curiosity, but they also play a geophysical role

> After that, we put another paragraph explaining why modelling giant dust particles requires to take intro account large-particle correction factor to the Stokes law, and how this has been done so far.
>
> When this is done, we present the goal of the paper, and its outline.
>
> In particular, we now present the precise goal of the papar towards teh end of the revised introduction, after giving more information on the context of the study and why we feel our work is useful in this context.

Lines 10-11: What is large-particle correction? Authors should introduce that term in order a less engaged reader can understand better the meaning of the sentence.

> With the reorganisation and the rewriting of the itroduction, the introduction of "large-particle correction" now comes later, and with a more detailed explanation of the term:
>
> "The sedimentation speed of giant particles deviates substantially from the Stokes law, an effect that can be taken into account using mathematical formulations known as large-particle corrections. Usually, these large-particle corrections are performed by using empirical formulations of the drag-coefficient $C_d$ as a function of the Reynolds number $Re$ (typically the one provided by [Clift and Gauvin, 1971]), and numerically solving an equation to obtain an estimate of the settling speed $v_\infty$ as a function of the characteristics of the particle and of ambient air." etc. etc.

Line 16: Please give the definition of giant particles.

    The definition of the giant mode for dust particles according to [Ryder et al., 2019] is now given in the second paragraph of the introduction.

Lines 16-17: References are needed here.

    In the revised version, the second paragraph in the introduction gives a bit more bibliography and context on giant dust particles. However, since this topic is not the heart of the manuscrip topic (but needed to justify why our study may beb useful), we did not want to increase too much the focus on this bibliographical field. At the end of this paragraph, we orient the reader towards some recent studies of the field for a more complete bibliography: "For a more complete bibliography, the reader is referred to van der Does et al. (2018), Ryder et al. (2019) and Drakaki et al. (2022)."

Lines 18-19: Why do we care about the missing from the models coarse dust particles. How do they affect the physical processes in the atmosphere?

    Some more arguments have been added to answer this question in the revised version: "The contribution of the giant mode is substantial, at least over the Sahara: Ryder et al. (2019) shows that not taking into account giant dust particles over the Sahara results in underestimating mass concentration by 40%, and extinction by as 18% for shortwave radiation and 26% for longwave radiation. Dust particles with diameter up to 100 µm are present not only above the Sahara (Ryder et al., 2019) but have also been observed, far away from emission sources.". As above, the reader could refer to the cited publications for more information on this point.

Line 26: The authors state that Drakaki et al. (2022) use Clift and Gauvin (1971) correction and performed the bisection method once for each model size bin. The word "once" is a little confusing, since viscosity depends on pressure and temperature, which changes at each time step and in each model grid box. Thus, the terminal velocity is calculated accordingly at each time step, in each model grid box and for each model size bin, adapting the bisection method. This makes the code even more time consuming.

    We are grateful to the Reviewer for this piece of information we had actually misunderstood. The precision brought by the Reviewer has been transcribed in the paper:

    "An exception to this is the recent development exposed by [Drakaki et al., 2022] in the GOCART-AFWA dust scheme of WRFV4.2.1. In that study, the [Clift and Gauvin, 1971] drag coefficient correction is taken into account by a bisection method, performed at each time step, in each model cell and for each model size bin to calculate the settling speed as a function of the particle properties and the atmospheric conditions."

Line 51: The expression for air viscosity is missing.

    It is now included (Eq. 4 in the revised version)

Line 60: For $Re < 0.1$ the consideration of free-slip correction should be added as it is described in Drakaki et al. (2022) and Mallios et al. (2020). Why did you omit it in both Stokes and Clift and Gauvin expressions? Could the

consideration of the free-slip correction possibly change the methodology? By not including the slip-free correction, makes the methodology valid only for $Re \geq 0.1$.

> We fully agree with this comment, and that this limitation restricted the use of our method greatly. Therefore, as described in the "General comments" section, an extension of our method to include the slip-correction term is now the object of Section 5, and the method is modified accordingly in the conclusion.

Line 61: Please describe in detail the iterative method you used. Also in line 68.

> The iterative method (which was actually only suggested in [van Boxel, 1998] but not described) is now described in the beginning of Section 6 of the revised manuscript.

Line 141: Please define the exact ranges that the expression is valid.

> We have added the following precision: "valid for all spherical particles with $D < 1000\,\mu$m and at least from the surface to $p = 200$ hPa (Figs. 3b and 4)"

Line 142: Can you provide an estimation of the computational benefit of the method?

> See the General comments section. Since both Reviewers indicated that this was missing, we have added Section 6 to perform this calculation.

Best regards,

The Authors.

**References**

[Clift and Gauvin, 1971] Clift, R. and Gauvin, W. H. (1971). Motion of entrained particles in gas streams. *The Canadian Journal of Chemical Engineering*, 49.

[Drakaki et al., 2022] Drakaki, E., Amiridis, V., Tsekeri, A., Gkikas, A., Proestakis, E., Mallios, S., Solomos, S., Spyrou, C., Marinou, E., Ryder, C. L., Bouris, D., and Katsafados, P. (2022). Modeling coarse and giant desert dust particles. *Atmospheric Chemistry and Physics*, 22(18):12727–12748.

[Ryder et al., 2019] Ryder, C. L., Highwood, E. J., Walser, A., Seibert, P., Philipp, A., and Weinzierl, B. (2019). Coarse and giant particles are ubiquitous in saharan dust export regions and are radiatively significant over the sahara. *Atmospheric Chemistry and Physics*, 19(24):15353–15376.

[van Boxel, 1998] van Boxel, J. (1998). Numerical model for the fall speed of raindrops in a rainfall simulator. Technical Report 1998/1, I. C. E. special report.